# Selection of Wastewater Treatment Technology: AHP Method in Multi-Criteria Decision Making

**Jasmina Ćetković [1,\*], Miloš Knežević [2], Radoje Vujadinović [3], Esad Tombarević [3] and Marija Grujić [4]**

1   Faculty of Economics Podgorica, University of Montenegro, 81000 Podgorica, Montenegro
2   Faculty of Civil Engineering, University of Montenegro, 81000 Podgorica, Montenegro
3   Faculty of Mechanical Engineering, University of Montenegro, 81000 Podgorica, Montenegro
4   Faculty of Civil Engineering, University of Belgrade, 11000 Belgrade, Serbia
*   Correspondence: jasmina@ucg.ac.me; Tel.: +382-67-652-016

**Abstract:** Wastewater treatment is a process that reduces pollution to those quantities and concentrations at which purified wastewater is no longer a threat to human and animal health and safety and does not cause unwanted changes in the environment. Municipal wastewater is classified as biodegradable water. Special importance should be given to wastewater with a high content of organic matter (COD), phosphorus (P) and nitrogen (N). MBBR technology, developed on the basis of the conventional activated sludge process and the bio filter process, does not take up much space and does not have problems with activated sludge, as in the case of conventional biological reactors, and has shown good results for the removal of organic matter, phosphorus and nitrogen. The aim of this paper is to optimize the wastewater treatment process in the municipality of Dojran, North Macedonia. Three alternative solutions for improving the capacity for wastewater treatment in the municipality of Dojran were analyzed. The shortlist of variants was made on the basis of several criteria, including: analysis of the system in the tourist season and beyond, assessment of the condition and efficiency of the existing wastewater treatment plant (WWTP) in combination with a new treatment plant, treatment efficiency when using different wastewater treatment technologies, the size of the site needed to accommodate the capacity, as well as the financial parameters for the proposed system. The selection of the most favorable solution for the improvement of the wastewater treatment system was made using the AHP (analytic hierarchy process) method. In order to select the optimal solution, a detailed analysis was conducted, considering several decision-making criteria, namely the initial investment, operating costs and management complexity. Based on the obtained results, Variant 3 was recommended, that is, the construction of a completely new station with MBBR technology, with a capacity for 6000 equivalent inhabitants.

**Keywords:** AHP method; MBBR technology; process optimization; wastewater treatment plant

## 1. Introduction

Choosing the optimal solution for wastewater treatment is a key stage in the optimization of the wastewater treatment process. This is because, according to some research, the world's population is expected to face the problem of water shortage if consumption remains at the current level [1]. Therefore, water reclamation and its reuse are the only possible solutions to this problem.

Wastewater treatment using moving bed biofilm reactor (MBBR) technology is used to filter wastewater in the industrial and municipal sectors. MBBR is the state-of-the-art wastewater treatment process that uses specialized biological technologies. This treatment can be used in the municipal and industrial sectors for nitrification, BOD removal and water purification, but it can also be integrated with other systems to achieve better results in pollutant removal. MBBR includes a simplified operating system that increases water purity beyond the conventional wastewater treatment limits. This biological treatment technology

is preferred over conventional approaches due to several comparative advantages, such as health and ecological advantages, high efficiency, convenience, small space requirements, cost effectiveness, flexibility and ease of operation, etc.

Before the advent of these reactors, many other types of conventional wastewater treatment systems were in use [2], which had significant disadvantages compared to MBBR. In response to those shortcomings, the moving bed biofilm reactor appeared in Norway in the late 1980s and early 1990s [3,4], as along with the first pioneering work on this technology [5]. The new MBBR has long been suggested to be suitable for dairy wastewater treatment [6], whereby the preference for the MBBR system was originally linked to the absence of sludge recycling and its ease of operation [7]. Meanwhile, in different parts of the world, different prominent variants of MBBR technology have been developed with the same basic principle [8,9]. The development of this technology was supposed to solve the problems of small communities regarding the need for small, easy-to-install wastewater treatment plants. Not long after, research reported the commercial success of MBBR technology in a significant number of countries around the world [10–13], which was followed by their even more extensive use in the wastewater treatment process [14]. In addition, some earlier studies examined the efficiency of upgrading an existing plant with MBBR technology in different climatic conditions and seasonal temperature fluctuations, and confirmed the justification of its use even at lower temperatures [15]. In the meantime, there was a need to increase the capacity of existing plants, which necessitated the further development of MBBR technology. Although MBBR technology has some drawbacks [16,17], it has numerous advantages compared to conventional biological treatments, such as space savings, improvement of performance and capacity with minimal additional costs, less clogging, the sludge does not require recirculation, the footprint is consistently reduced, increased biofilm resistance to temperatures, shock loads, toxic compounds, pH, etc. [18–20]. Thanks to its simplicity, flexibility, robustness and compactness, MBBR technology has seen a growth of its application [21] and has become a widely recognized technology for wastewater treatment [22], where it has shown enormous potential in reducing the load of contamination and pollution of municipal and industrial waters [23–27]. The advantages of MBBR technology have been demonstrated in agriculture, the denitrification of drinking water [11,28], and oil refinery wastewater [29], as well as in hospital wastewater treatment [30]. In addition, with small modifications, it is possible to adapt the existing infrastructure to host MBBR [31]. Such high-performance capability in carbon and nitrogen removal and a compact footprint enable MBBR technology to be a good solution for either small decentralized facilities or for upgrading existing centralized facilities [32].

Appreciating the fact that finding suitable technologies for efficient wastewater treatment and its reuse is important for the sustainability of the industry [33–35], a number of studies examined the use of MBBR in specific industries [36–40], but also compared the economic and environmental advantages and disadvantages of using different biological methods in the treatment of industrial wastewater, suggesting that the application of MBBR in some industries has advantages over other methods and technologies [41]. In one of the most recent studies, the outstanding results of the application of MBBR technology for municipal and industrial wastewater were highlighted and it was confirmed that the maximum removal efficiencies are BOD of 97%, COD of 96%, phosphorus of 99% and oxygen of 99%, at an HRT of 2–6 h [42]. At the same time, it is a technology that is applicable for a wide range of wastewater flows, from 10 to 150 thousand $m^3 day^{-1}$ [43], therefore, various mathematical methods were developed for calculating the reactor volume, organic effluent concentration and substrate removal rate [44]. Kawan et al. [45] pointed out the advantages of MBBR technology as a highly modular system that can be used for very low or very high concentrations, as well as for polishing, therefore techno-economic analyses are necessary. Khudhair et al. [46] indicated in their recent research the problem of excess sludge production, which represents one of the limitations of the biological activated sludge process, which can be overcome by upgrading the MBBR process. Namely, their analysis showed that the variant of upgrading the MBBR process to the integrated fixed-film

activated sludge (IFAS) completely eliminates sludge and that the system achieved low effluent pollutants concentrations.

One of the key advantages of MBBR technology is its high level of treatment efficiency. Parivallal et al. [47] pointed out in their recent study the high efficiency of MBBR technology in wastewater treatment processes. Namely, a treatment plant with a capacity of one million liters per day meets all standards in terms of basic water quality parameters, such as BOD, COD, TKN and TSS. Similar results, regarding high treatment efficiency, were confirmed by Masłoń and Tomaszek [48], which involved a 15 L-laboratory scale MBSBBR (moving bed sequencing batch biofilm reactor) model. The results of this study indicate a high level of average efficiency in removing COD, total nitrogen (TN) and total phosphorus (TP), ranging from 97.7 $\pm$ 0.5%, 87.8 $\pm$ 2.6% and 94.3 $\pm$ 1.3%, respectively, while the nitrification efficiency reached a level in the range of 96.5–99.7%. Zhou et al. [49] confirmed the feasibility of the two-stage anoxic/oxic moving bed biofilm reactor (TS-A/O-MBBR) in a full-scale municipal wastewater treatment plant (WWTP), as well as its high efficiency and ability to meet high standards in the biological nitrogen removal process. The maximum removal efficiency of total nitrogen and the minimum concentration of total nitrogen in the effluent reached 91.76% and 4.12 mg/L, respectively. Czarnota and Masłoń [50] showed that not all MBBR systems are equally efficient. Their study aimed to assess the effectiveness of MBBR reactors with EvU-Perl carriers. The results of this study, which was aimed at improving efficiency, indicate the need for better sludge management, increasing the volume of nitrification chambers or replacing the biofilm carrier, as annual analyses showed a decrease in biogenic compounds below the prescribed level.

The treatment efficiency and cost-effectiveness of MBBR technology primarily depend on the type of the biofilm carriers that can be modified according to the process, which is the main advantage of this technology. For example, Chu and Wang [51] compared the efficiency between two different biofilm carriers (polymer polycaprolactone—PCL and inert poly-urethane foam—PUF) and gave preference to PCL carriers with a low C/N ratio in terms of TN removal. In a recent study, Ashkanani et al. [52], using MBBR with three AnoxKaldnes media, determined the influence of the shape and surface of biocarriers on efficiency, favoring a biocarrier that has a smaller specific surface due to less clogging. Additionally, Maziotti et al. [53] preferred AnoxKaldnes K3 over Mutag BioChip in their study due to higher COD removal efficiency. Shitu et al. [54] concluded that novel sponge biocarriers (SB) in MBBR increase the diversity of the functional microbial communities and achieve the highest nitrification performance.

Nevertheless, regardless of the wide practical use of MBBR technology, a review of the literature reveals that the research of this technology is limited compared to the literature focusing on conventional systems [18].

Sequencing batch reactors (SBR) are a variation of activated sludge and have been widely applied in wastewater treatment for almost a century due to their unique advantages [55]. As environmental standards became stricter and the number of new pollutants grew exponentially, SBR technology, as a modification of the popular activated sludge process (ASP), gained in importance and application. From small community use to the treatment of high-hardness industrial wastes, the use of SBRs has expanded to the biological treatment of industrial waters containing organic chemicals that are difficult to remove. As one of the integrated systems for anaerobic–aerobic bioreactors, SBR processes are often used in industrial wastewater treatment due to their compactness and high efficiency [56–58], and are used less often for domestic wastewater treatment [59]. The advantages of this technology are not only the high performance in low or varying flow patterns, but also the lower costs over a longer period of time. Applying technology SBR in municipal wastewater treatment with biological process nitrification/denitrification is a major opportunity and a good chance for developing countries to reach sustainable development and ecological balance in their urban areas. Therefore, Quan and Gogina [60] indicated more stable efficiency in the removal of pollutants and decreased environmental damage by 8–11 times while achieving optimal operation by bio-film in their study on

the technical economic efficiency of SBR. Dutta and Sarcar [61] emphasized that SBR technologies save more than 60% of the operating expenses for a conventional ASP, and high effluent quality is achieved in a very short aeration time. Ćetković et al. [62] considered the financial and socio-economic feasibility of SBR as one of the variant solutions in the CBA implemented. Due to their excellent process control capabilities and operational flexibility, SBRs are widely used to treat wastewater, but future research on SBR control strategies and the development of intelligent control systems can make SBRs more adaptable to changing environmental conditions and changing wastewater quality in order to maintained optimal and reliable effluent quality. Alagha et al. [63] investigated the performance of a pilot-scale SBR process for the treatment of municipal wastewater quality parameters in terms of two scenarios, namely, pre-anoxic denitrification and a post-anoxic denitrification scenario. Their results confirm that the post-anoxic denitrification scenario was more efficient for higher qualify effluent, which is why the suitability of using this technology in remote areas in arid regions with a high reusability potential is suggested. Fernandes et al. [64] analyzed the microbial diversity and performance in the SBR of a decentralized full-scale system for urban wastewater treatment under limited aeration and confirmed the viability and efficiency of the reactor to treat domestic wastewater. Numerous papers have shown that SBR use results in a more efficient process that requires less energy consumption than conventional systems [65–67].

The aim of this paper is to optimize the wastewater treatment process in the municipality of Dojran, North Macedonia, i.e., to choose between the three offered variant solutions. It was created as a result of the author's involvement in the preparation of a feasibility study related to the improvement of the wastewater treatment system in this municipality, within a wider project financed by SIDA, entitled "Building Municipal Capacity for Project Implementation". The optimization of the wastewater treatment process in the municipality of Dojran should contribute to the implementation process of European standards for environmental protection [68,69] to ensure that the maximum allowed concentrations of pollutants in the wastewater discharged into the recipient are not exceeded. This study analyzes several proven wastewater treatment technologies that are applicable to the location in question.

The article is organized into several sections. In the Introduction, certain aspects of the application of MBBR technology in the wastewater treatment process have been articulated through a review of the relevant literature. The second section of the paper presents the current situation and problems regarding the wastewater treatment process in the municipality of Dojran, as well as the methodology we used to select the optimal variant of wastewater treatment technology in Nov Dojran. In addition, in this section, we determine the equivalent inhabitants and the amount of wastewater that will be generated in the municipality of Dojran as key input for the analysis in the paper. In the third section of the paper, the analysis of three technical variant solutions related to the problem of improving the capacity for wastewater treatment for the municipality of Dojran is presented. In the fourth section, the selection of the optimal variant of wastewater treatment in the municipality of Dojran is made using the AHP (analytic hierarchy process) method. The last section provides a concluding summary of the research, points out the limitations of the approach and suggests ideas for future research on this topic.

## 2. Data and Methodology

The first point of this section is to provide an overview of the situation and challenges in the wastewater treatment process in the municipality of Dojran, North Macedonia. The second part briefly presents the basics of the relevant methodology that we used in order to select the optimal variant of wastewater treatment process in Nov Dojran. In the third part, we determine the equivalent inhabitants and the amount of wastewater that will be generated in the municipality of Dojran as basic input that is necessary for the selection of the optimal variant of wastewater treatment technology in this municipality.

*2.1. Overview of the Situation and Challenges in the Wastewater Treatment Process: Municipality of Dojran, North Macedonia*

The hydrography of the municipality of Dojran mainly consists of the Dojran Lake, smaller springs and streams, as well as a few artificial reservoirs. Dojran Lake is located at a height of 140 m above sea level. The surface area of the lake is 42.5 km$^2$, of which 26.58 km$^2$, or 62.54% belongs to the Republic of North Macedonia, and 15.92 km$^2$, or 37.46% to Greece. The water volume of the lake is 289.61 million m$^3$. The length of the lake is 8.9 km, and the greatest width is 7.1 km. The average depth is 6.7 m, and the greatest is about 10 m. In the period from 1988 to 2000, the level of the lake water constantly decreased, reaching the lowest point of −3.88 m below the zero point. The hydrography of Dojran Lake has been significantly enhanced in recent years by the construction of the Gavoto–Dojran canal, which brings additional water into Dojran Lake, increasing the lake level by about 1.80 m from the absolute minimum.

According to the data from the last official population census from 2002 [70], the municipality of Dojran has 3426 inhabitants, of which Nov Dojran has 1100 inhabitants. The company JPKD Komunalec—Polin Star Dojran was established to provide utility services in the area of the municipality of Dojran. In the context of all the services it offers and performs, of special interest are the services of water collection, treatment and supply, wastewater disposal, construction of water supply systems, construction of sewerage systems and construction of a storm water drainage system.

Thus far, the settlements of Star Dojran, Nov Dojran and Sretenovo have access to the sewerage system. The network is divided into main (primary) and secondary systems. The secondary sewerage network in the municipality of Dojran has a total length of 7650 m, with a diameter ranging from Ø 150 mm to Ø 250 mm. It covers the aforementioned settlements and serves to collect wastewater from households and transport it to the main collector. The main collector system was built in 1989 and stretches along the entire length of Dojran Lake on the Macedonian side. The collector is 8340 m long, made of PVC pipe, with a diameter ranging from Ø 250 mm to Ø 500 mm. Submersible fecal pumps installed in ten pumping stations arranged along the length of the collector pump fecal water to WWTP Toplec in Nov Dojran.

The existing WWTP Toplec is located in the suburb of Nov Dojran and represents the completion of the sewerage system in the municipality of Dojran. The process of wastewater disposal ends with a treatment plant from which the treated water is discharged into Dorjan Lake. It was built in 1988 as the last point of the eastern and western collectors. The plant was designed for 8000 equivalent inhabitants and consists of two blocks, the first of which is technically outdated and out of use, while the second block is in operation. A project for the reconstruction of the second block was prepared in order to increase the efficiency of WWTP Toplec by replacing and supplementing the treatment technology. However, sludge treatment would be a problem in the functioning of the plant even after reconstruction. Sludge dewatering is not foreseen, and the sludge is often left to dry in fields. A new technological solution should overcome this problem. In addition, the storm sewer system is not fully developed and covers only a small part of the municipality. Reconstruction of the existing system or construction of a new WWTP should improve the quality of surface and ground water and soil in the wider region. However, there are also possible negative impacts of reconstruction or construction of a new WWTP, as well as from the purification of wastewater during the so-called operational phase. It is expected that reconstruction or new construction will mainly result in waste that is not classified as hazardous (in accordance with the waste management regulation), such as stones, mixed municipal waste, etc. All waste should be disposed of in landfills. The possible amount of hazardous waste will be small.

The composition of wastewater, which should be treated in the planned WWTP in the municipality of Dojran, corresponds to the typical composition of wastewater. It is necessary to ensure the quality of the effluent that is discharged into Dojran Lake in

accordance with the standards of the EU Directive for urban wastewater. According to the local regulation [71], Dojran Lake is classified in the II (second) category.

During the exploitation process, that is, the operational phase of the WWTP, several types of waste will be generated, which can be classified into two main types: waste resulting from the wastewater treatment process and waste resulting from the maintenance of the WWTP itself. Other phenomena that could disrupt the comfort of citizens are noise during the period of construction activities and unpleasant odors during the exploitation process. The realization of the project itself will have a positive impact on the environment because the long-standing problem of loading Dojran Lake with organic matter originating from municipal wastewaters will be solved.

*2.2. AHP Method*

To select the optimal technology for wastewater treatment in Nov Dojran, the analytic hierarchy process (AHP) method was chosen as one of the most frequently used multi-criteria decision-making methods [72,73], which is used when making decisions in complex problems. It is used with a multi-layered hierarchical structure of goals (which we want to achieve), criteria, sub-criteria and alternatives that we consider. The input data were derived through several comparisons. These comparisons were used to define the degree of importance of the criteria used in decision-making, as well as to determine the relative measures for evaluating alternatives according to each separate decision criterion. The method, which is based on a mathematical, but also human approach, deconstructs the problem by hierarchy and enables evaluation according to different criteria. The AHP method includes four main steps:

- Development of a hierarchy of interconnected decision elements that describe the problem;
- Comparing pairs of decision elements, usually using a 1–9 comparison scale, to obtain input data;
- Calculation of relative weightings of decision-making elements, most often using the method of characteristic values;
- Aggregation of relative weightings of decision elements in order to calculate the rating of alternative decision possibilities.

The relative importance of criteria *i* and *j* is evaluated with values from 1 to 9 [74–76]. The significance of those values is presented in Table 1.

**Table 1.** Table of coefficients of importance of criteria according to Saaty [77].

| Intensity of Importance | Definition | Explanation |
| --- | --- | --- |
| 1 | Equal importance | Criteria *i* and *j* are equally important |
| 3 | A moderate advantage of one criteria over another | Criterion *i* is moderately more important than *j* |
| 5 | Essential, strong importance | Criterion *i* is significantly more important than *j* |
| 7 | Very strong importance | Criterion *i* is very significantly more important than *j* |
| 9 | Extreme importance | Criterion *i* is extremely more important than *j* |
| 2, 4, 6, 8 | Mean values between two adjacent estimates | To define the rating, a comparison of two estimates is needed (a compromise is needed) |
| Reciprocities | If one activity has one of the numbers above (e.g., 3) compared to a second activity, then the second activity has a reciprocal value (i.e., 1/3) when compared to the first. | |
| Rationality | Coefficients resulting from forcing consistency of estimation | |

*2.3. Determining the Equivalent Inhabitants and the Amount of Wastewater*

The wastewater collection and disposal system in the municipality of Dojran covers settlements along Dojran Riviera, namely Nov Dojran, Star Dojran and Sretenovo. The system consists of a secondary and primary sewage network, which ends with the WWTP. For other settlements in the municipality, the construction of a fecal sewage network with a

small WWTP is planned, which is why these settlements are not included in the calculation for determining the equivalent inhabitants, that is, the amount of wastewater.

Considering that Dojran is a tourist center with the peak season in the summer months (June, July and August), two analyses of the equivalent inhabitants were carried out: out-of-season (with only the permanent population included) and in-seasonwith permanent population and tourists). Determining the equivalent inhabitants out-of-season is presented in Table 2.

**Table 2.** Determining the equivalent inhabitants for the municipality of Dojran—out-of-season.

| | | |
|---|---|---|
| 1. | Population according to 2002 census [70]<br>Star Dojran<br>Nov Dojran<br>Sretenovo<br>Total population in the area of interest | $N_1 = 363.00$ inhabitants<br>$N_2 = 1100.00$ inhabitants<br>$N_3 = 315.00$ inhabitants<br>$N_{(2002)} = 1778.00$ inhabitants |
| 2 | Population at the end of exploitation period [78] | $N_k = 1908.37$ inhabitants |
| | $N_k = N_0(1 + p/100)^n$ | $N_k = 2000.00$ inhabitants |
| | $N_0$—current population | $N_0 = 1833.60$ inhabitants |
| | $p$—population growth | $p = 0.16\%$ |
| | $n$—exploitation period | $n = 25.00$years |
| 3. | Standard for water supply | $Q_0 = 150.00$ l/day/person |
| 4. | Standard for sewerage | $Q_k = 150.00$ l/day/person |
| 5. | Average wastewater emission per day<br>$Q_{av/day} = \frac{Q_k \cdot N_k}{1000}$ | $Q_{av/day} = 300.00$ m$^3$/day |
| 6. | Maximum wastewater emission per day<br>$Q_{max/day} = a_1 Q_{av/day}$<br>$a_1$—maximum daily uneven distribution coefficient | $Q_{max/day} = 450.00$ m$^3$/day<br>$a_1 = 1.50$ |
| 7. | Average wastewater emission per hour<br>$Q_{av/h} = \frac{Q_{max/day}}{24}$<br>$q_{av/sec} = \frac{Q_{av/h}}{3.6}$ | $Q_{av/h} = 18.75$ m$^3$/h<br>$q_{av/sec} = 5.21$ l/sec |
| 8. | Maximum wastewater emission per hour<br>$Q_{max/h} = a_2 Q_{av/h}$<br>$a_2$—maximum daily uneven distribution coefficient<br>$q_{max/sec} = \frac{Q_{max/h}}{3.6}$ | $Q_{max/h} = 30.00$ m$^3$/h<br>$a_2 = 1.60$<br>$q_{max/sec} = 8.33$ l/sec |

The wastewater plant is sized for the average wastewater emission per hour, with the possibility of maximum wastewater emission per hour. We should emphasize that the infiltration of water of another origin (e.g., storm water, lake water) is not included in the calculation because it significantly increases the amount of water for purification, which increases the cost of the plant while reducing efficiency.

In order to determine the equivalent inhabitants during the tourist season, that is, the amount of wastewater, data on the hospitality industry is needed, primarily on its nature and capacities [79]. These data are shown in Table 3. The defined growth of seasonal visitors and the hospitality industry is 3% until 2029 and 0.5% in the remaining period [78].

The number of equivalent inhabitants during the tourist season is determined based on the character and capacity of the hospitality industry for the municipality of Dojran (Table 4).

**Table 3.** Data on the hospitality industry in the municipality of Dojran.

| Year | 2016 | 2029 | 2046 | Year | 2016 | 2029 | 2046 |
|---|---|---|---|---|---|---|---|
| Total catering facilities: | 14 | 20 | 22 | Capacity/number of places: | 483 | 710 | 772 |
| - Restaurants | 6 | 8 | 9 | - Restaurants | 193 | 284 | 309 |
| - Fast food | 3 | 4 | 4 | - Fast food | 97 | 142 | 154 |
| - Dairy restaurants | 1 | 2 | 2 | - Dairy restaurants | 48 | 71 | 77 |
| - Coffee bars | 4 | 6 | 7 | - Coffee bars | 145 | 213 | 232 |
| Total hospitality facility with the possibility of an overnight stay: | 47 | 68 | 73 | Capacity/number of beds: | 2079 | 3053 | 3323 |
| - Hotels | 23 | 34 | 37 | - Hotels | 1040 | 1527 | 1662 |
| - Resorts | 12 | 17 | 18 | - Resorts | 520 | 763 | 831 |
| - Other type | 12 | 17 | 18 | - Other type | 520 | 763 | 831 |

**Table 4.** Determining the equivalent inhabitants for the municipality of Dojran—in-season.

| 1. Determining the equivalent inhabitants | | | |
|---|---|---|---|
| Description | Number of visitors | Standard | Total amount |
| / | / | l/day/person | l/day |
| Restaurants | 309 | 100 | 30,882.62 |
| Fast food | 154 | 10 | 1544.131 |
| Dairy restaurants | 77 | 10 | 772.0654 |
| Coffee bars | 232 | 10 | 2316.196 |
| Hotels | 1662 | 200 | 332,323.8 |
| Resorts | 831 | 120 | 99,697.14 |
| Other types of accomodation | 831 | 120 | 99,697.14 |
| Total amount of wastewater | | | 567,233.1 |
| Average drainage rate | | | 150 |
| Equivalent inhabitants—seasonal visitors | | | 3781.554 |
| Equivalent inhabitants—everyday visitors | | | 1908.37 |
| Total equivalent inhabitants in season | | | 5689.92 |
| Determined equivalent number of inhabitants in the season | | | 6000 |

| 2. Average wastewater emission per day | |
|---|---|
| $Q_{av/day} = \frac{Q_k \cdot N_k}{1000}$ | $Q_{av/day} = 900 \text{ m}^3/\text{day}$ |

| 3. Maximum wastewater emission per day | |
|---|---|
| $Q_{max/day} = a_1 Q_{av/day}$ <br> $a_1$—maximum daily uneven distribution coefficient | $Q_{max/day} = 1350 \text{ m}^3/\text{day}$ <br> $a_1 = 1.50$ |

| 4. Average wastewater emission per hour | |
|---|---|
| $Q_{av/h} = \frac{Q_{max/day}}{24}$ | $Q_{av/h} = 56.25 \text{ m}^3/\text{h}$ |
| $q_{av/sec} = \frac{Q_{av/h}}{3.6}$ | $q_{av/sec} = 15.63 \text{ l/sec}$ |

| 5. Maximum wastewater emission per hour | |
|---|---|
| $Q_{max/h} = a_2 Q_{av/h}$ | $Q_{max/h} = 90.00 \text{ m}^3/\text{h}$ |
| $a_2$—maximum daily uneven distribution coefficient | $a_2 = 1.60$ |
| $q_{max/sec} = \frac{Q_{max/h}}{3.6}$ | $q_{max/sec} = 25 \text{ l/sec}$ |

## 3. Analysis: Variant Solutions for Improving the Capacity for Wastewater Treatment

Three technical alternative solutions related to the problem of improving wastewater treatment capacity for the municipality of Dojran were analyzed. The short list of variants was made on the basis of several criteria, including system analysis (in and out of the tourist season), assessment of the condition and efficiency of the existing WWTP in combination with a new treatment plant, treatment efficiency when using different wastewater treatment technologies, size of the site required to accommodate the treatment capacity and

financial parameters for the proposed system, i.e., the initial investment and the necessary maintenance budget.

As stated in Section 2, the main reconstruction project of the second block of the existing WWTP was carried out, which provided for an increase in efficiency and capacity up to 6000 equivalent inhabitants. In addition, there is a marked increase in the number of equivalent inhabitants during the tourist season (6000 equivalent inhabitants, compared to 2000 out of season). Because of this, the investor insisted that two options should be considered, which are based on the planned reconstruction of the existing WWTP, which would be used during the season for 6000 equivalent inhabitants. In both variants, the construction of new systems is foreseen—MBBR and SBR, which would operate out of season with a capacity adjusted for 2000 equivalent inhabitants. In the third variant, the existing WWTP is provided as a reserve capacity that can be reconstructed if there is an increase in need.

The solutions that are applicable for the given conditions are as follows:

1. Exploitation of the existing WWTP in accordance with the main project for the reconstruction and construction of the new MBBR wastewater treatment system for the calculated equivalent inhabitants of Dorjan, which will be used outside the tourist season—Variant 1;
2. Exploitation of the existing WWTP in accordance with the main project for reconstruction during the tourist season and construction of a new SBR (sequencing batch reactor) for the calculated equivalent inhabitants of Dojran, which will be used outside the tourist season—Variant 2;
3. Construction of a new MBBR wastewater treatment system for 6000 equivalent inhabitants with two modules, of which, module two will be active in the tourist season, while module one will be active only outside the tourist season—Variant 3.

### 3.1. Variant 1—Combination of the Existing WWTP and the New MBBR Wastewater Treatment System for 2000 Equivalent Inhabitants

As the calculations of the equivalent inhabitants confirm, due to the fact that Dojran is a tourist center, there are large variations in wastewater emissions during the year. Therefore, in order to achieve greater efficiency and economy, it is planned to build a new plant that would operate throughout the year and serve the permanent residents of the municipality of Dojran, while during the tourist season, in accordance with the reconstruction project, the existing WWTP would also be activated [58]. Variant 1 is presented in Figure 1.

The new treatment plant would be located next to the existing one. It is envisaged that the water will be directed to the distribution shaft equipped with valves via the last pumping station of the main collector system, from where one line would lead to the existing WWTP and the other line would lead to the new plant. The area (approx. 660.0 m$^2$) where the new treatment plant is located is part of the private plot KP 295 and should be subject to expropriation.

The constituent elements of the planned WWTP are separated into common facilities and equipment and, as such, are most often found in this type of plant. Thus, the following elements are planned for the MBBR treatment station for 2000 equivalent inhabitants:

○ Distribution shaft with a sluice gate for directing water to the wastewater treatment plant;
○ Inlet pump station—reinforced concrete facility with an automatic coarse screen for bulky waste and a panel house mounted on a steel structure above the pump station;
○ Flow measuring shaft for measuring the flowrate of wastewater—reinforced concrete facility with a built-in electromagnetic flow meter and the necessary equipment;
○ Equalization pool—reinforced concrete facility with a compressor station and built-in air distribution system, diffusers and mixers;
○ Modular (assembly–disassembly) container plant, two stage MBBR–BNB bioreactor with a moving bed, in which there is an automatic mixer, a fine screen and a sludge pump;

○ A modular container that houses a tank (assembly–disassembly) with a built-in cartridge for the microfiltration of treated water, a pre-pumping station and another part of the control equipment (PLC system) with a voltage regulator for the entire system;

○ Sludge storage tank (assembly–disassembly) with a compressor unit;

○ Emergency shaft/reinforced concrete facility made from ready-made prefabricated elements;

○ Collecting shaft with a channel for purified water discharge to the recipient/reinforced concrete facility—from ready-made prefabricated elements;

○ Press and dryer—for sludge dewatering and drying.

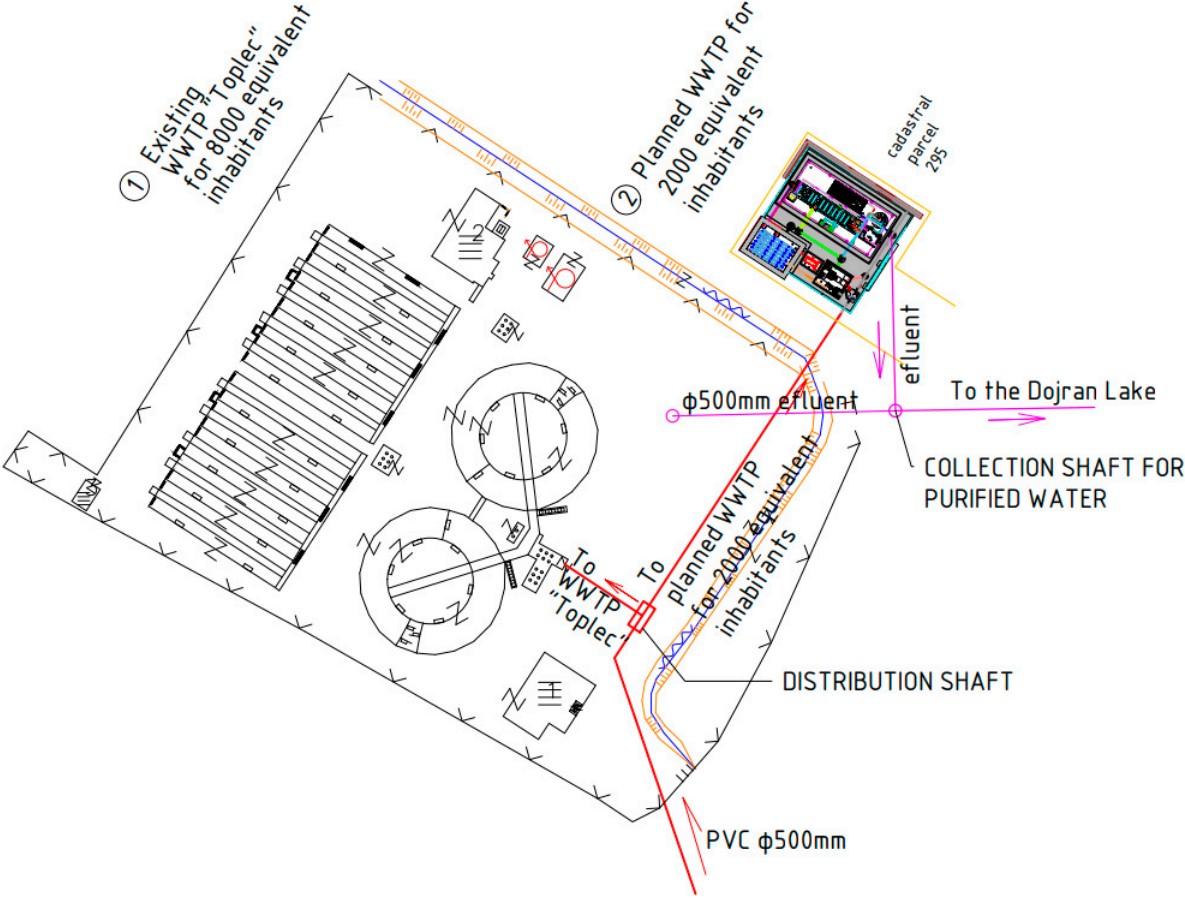

**Figure 1.** Combination of existing WWTP and new MBBR.

Considering a high level of removal (Table 5), effluent from plants can be discharged into natural streams, as it corresponds to all quality standards.

**Table 5.** Efficiency of MBBR as a function of filing.

| Treatment Efficiency | gBOD5/m$^2$day |
|---|---|
| 75–80% | 20.0 |
| 80–85% | 10.0 |
| 85–90% | 6.0 |
| 90–95% | 4.5 |
| 95%–100% | 2.5 |

MBBR technology for wastewater treatment has significant advantages. It enables a long retention time for activated sludge, which is good for nitrification. The process can be carried out without a secondary precipitator. Sediment production is reduced. MBBR requires a small area, while the capacity/space ratio of the plant is maximized. It achieves

high flexibility in operation in the range of carrier fill from 30 to 70%. A two-stage biological process (high and medium load) increases efficiency and adapts to variable raw water inflow. The carrier material cannot be damaged (there are plants that are up to 20 years old and still use the same carriers). The thickness of the biofilm is controlled and maintained by the continuous separation that occurs under the action of aeration and mixing.

This technology is characterized by a smaller number of disadvantages. Since it is a biological process, the operation requires professional personnel to operate the plant. In addition, it also requires the engagement of qualified operators to ensure that there are no losses.

For the analyzed wastewater treatment technology, the cost calculation of the wastewater treatment plant was made, which included construction and craft work, procurement and installation of the equipment, connection of the plant to the distribution network, as well as its commissioning. The investment costs for this type of wastewater treatment technology were determined on the basis of several previously designed and constructed plants of this type (e.g., WWTP for S. Jurumleri, municipality Gazi Baba for 3500 equivalent inhabitants; WWTP for Novo Konjarevo, municipality Novo Selo for 1000 equivalent inhabitants; WWTP Ilovica and Štuku, municipality Bosilovo; WWTP Stračinci, municipality Gazi Baba). The analysis includes the calculation of plant maintenance and management costs. In accordance with the recommendations of the equipment manufacturer, as well as previous positive engineering practices, the annual cost calculation includes electricity costs (under normal operating conditions of the plant), employee costs (salaries and other related expenses), ongoing equipment maintenance and the servicing and cleaning of plant parts.

According to the current market prices, the investment cost for the construction of a wastewater treatment plant for 2000 equivalent inhabitants with MBBR and all the necessary stages of wastewater treatment is estimated at EUR 1,450,500. In addition, EUR 64,750 should be provided for operating costs on an annual basis. Given that, in this variant, the operation of the existing WWTP is planned during the tourist season, it is necessary to include the costs of its reconstruction, as well as its operating costs in the analysis. They amount to EUR 2,000,000 for reconstruction and EUR 101,500 per year for operating costs. Therefore, for Variant 1, the total investment costs amount to EUR 3,450,500, and the total operating costs are EUR 166,250.

According to the construction dynamics plan (Table 6), the construction would take place in four phases: construction of the MBBR for 2000 equivalent inhabitants with all the elements necessary for pretreatment and biological treatment (phase 1), reconstruction of the existing WWTP (phase 2), construction of the press (phase 3) and construction of a dryer (phase 4).

*3.2. Variant 2—Combination of Existing WWTP and New SBR Wastewater Treatment Systems for 2000 Equivalent Inhabitants*

The second variant includes the utilization of the existing WWTP (in accordance with the main reconstruction project) during the tourist season, and the construction of a new SBR wastewater treatment system for the calculated equivalent inhabitants of Dojran, which will be used outside the tourist season.

The new treatment plant would be located next to the existing one, as presented in Figure 2. It is envisaged that the water will be directed to the distribution shaft equipped with valves via the last pumping station of the main collector system, from which one line would lead to the existing WWTP and the other line would lead to the new station. The area (about 900 m$^2$) where the new treatment station is installed is part of the private plot KP295, and it should be subject to expropriation.

**Table 6.** Construction of new MBBR for 2000 equivalent inhabitants and reconstruction of the existing WWTP.

| MBBR technology for 2000 equivalent inhabitants | | |
|---|---|---|
| Dynamics of construction by phases | Costs | Amount (excluding VAT) |
| / | Investment costs | [EUR] |
| Phase 1 | Construction of a WWTP with all the elements required for pre-treatment and biological treatment | 800,500 |
| Phase 3 | Construction of a sludge dewatering press | 300,000 |
| Phase 4 | Construction of a dryer for drying sludge | 350,000 |
| | Σ | 1,450,500 |
| | Operating costs | [EUR/year] |
| | Electricity consumption - On average, 0.83 kWh/m$^3$ of purified water and for regular equipment service | 10,400 |
| | Ongoing service staff | 6000 |
| | Maintenance of the dryer—on average, 23.77 kWh/ton of sludge produced | 47,110 |
| | Maintenance of the press—on average, 12.5 kWh (the press would operate 2–3 h per day) | 1240 |
| | Σ | 64,750 |
| Existing WWTP "Toplec" | | |
| Dynamics of construction by phases | Costs | Amount (excluding VAT) |
| | Investment costs | [EUR] |
| Phase 2 | Reconstruction of the station | 2,000,000 |
| | Σ | 2,000,000 |
| | Operating costs | [EUR/year] |
| | Electricity consumption and regular equipment servicing | 71,500 |
| | Ongoing service staff | 30,000 |
| | Σ | 101,500 |

The elements of the planned WWTP are separated into common facilities and equipment that mainly occur for this type of technology. The SBR wastewater treatment system for 2000 equivalent inhabitants includes the following elements: inlet shaft, pumping station with fine screen, flow meter, retention basin, grease and oil trap, biological reactor (aeration and phosphorus elimination), outlet flow meter, clarifier, sludge dewatering press, sludge dryer and service facility.

In an SBR wastewater treatment system, the technological process includes pretreatment (removal of coarse and fine particles, grease and sand), a secondary process (elimination of carbon compounds COD, BOD5, elimination of ammonium) and a tertiary process (dephosphatization, chemical filter) and sludge line (thickening, dewatering/dehydration and drying).

The purification process is divided into two lines: primary (wastewater line) and secondary (sludge line). Wastewater from the municipality of Dojran (to be purified in the new WTTP) flows from the separation shaft directly into the mechanical purification plant, where wastewater is purified from solids. The water is pumped into the reservoir where grease and oils are separated. Wastewater from the grease and oil separator is pumped into the SBR reactors. Air is directly added to the SBR reactor via a blower. The chemical destruction of phosphorus carried out in the SBR reactors is followed by the

sludge disposal stage. The excess sludge is pumped into the sludge tank, after which the purified wastewater is discharged by gravity into the well and then into the receiver. Excess sludge from the wastewater treatment process is stored in a tank that is gravity-connected to the detention basin, so that sludge from the top of the tank will overflow into the detention basin. The concentrated activated sludge from the tank will be transported to the press and then to the dryer.

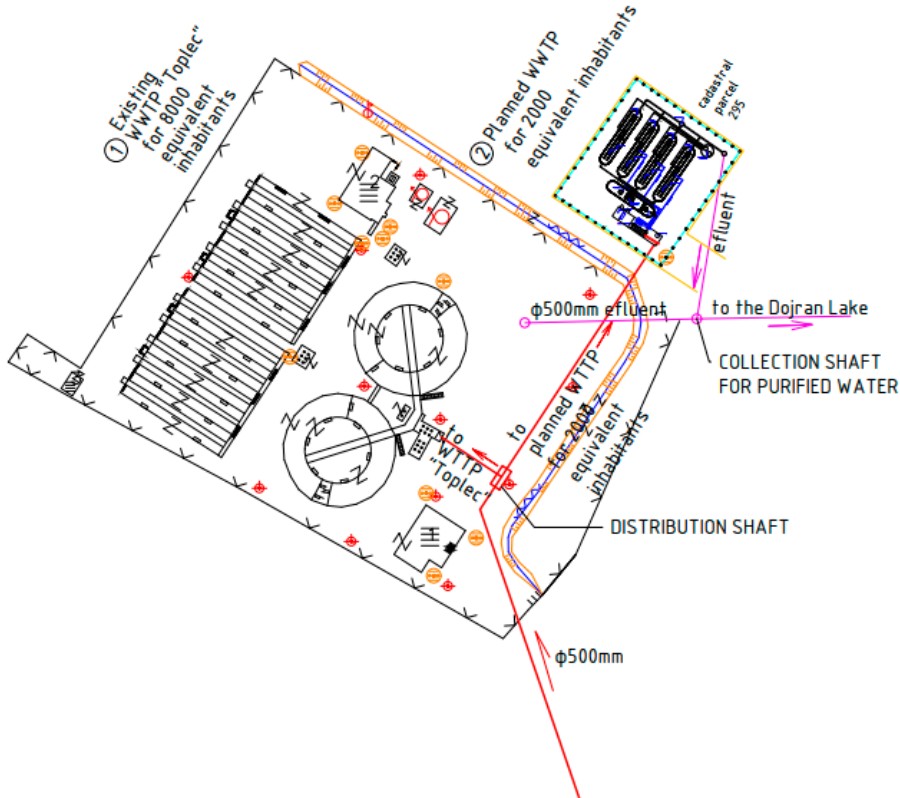

**Figure 2.** Combination of the existing WWTP and a new SBR.

The application of the SBR system is characterized by certain advantages, but also some disadvantages. One of the advantages is that the levelling of the basin and the primary clarifier (in most cases) can be achieved in one reactor. Biological treatment and secondary sedimentation can be achieved in one reactor. Flexibility and process control is ensured. The SBR system occupies a small area, and capital investment savings are achieved by eliminating the sedimentation tank and other equipment. However, the SBR system has its drawbacks. It requires a higher level of sophistication (compared to conventional systems), especially for larger systems. In addition, it requires a higher level of maintenance (compared to conventional systems), associated with more sophisticated controls, automatic switches and automatic valves. In the decanting phase, it is necessary to avoid capturing floating substances from the water. Depending on the aeration system used by the manufacturer, it may be necessary to include aerobic devices during the selected management cycle. Additionally, a sedimentation basin may be required after the SBR process, depending on the downstream processes.

For the analyzed wastewater treatment technology, the cost for the complete construction of the station with all the construction and craft work, procurement and installation of the equipment, and connection of the station to the electrical distribution network, as well as its commissioning was calculated. The investment costs for this type of wastewater treatment technology were determined on the basis of several previously designed and constructed stations of this type (WWTP for the village of Mavrovi Anovi, municipality of Mavrovo and Rostuša; WWTP for Millennium Cross for 1000 equivalent inhabitants with SBR technology; WWTP for the village of Stenje for 900 equivalent inhabitants with SBR

technology). The analysis also includes the calculation of station maintenance and management costs. In accordance with the recommendations of the equipment manufacturer, as well as previous positive engineering practices, the following annual costs are included in the calculation: electricity costs under normal operating conditions of the station, employee costs (salaries and other related expenses), ongoing equipment maintenance and servicing and cleaning of station parts.

According to the current market prices, the investment cost for the construction of a wastewater treatment plant for 2000 equivalent inhabitants with an SBR system and all the necessary stages for wastewater treatment is estimated at EUR 1,456,000 (VAT excluded). Additionally, EUR 69,150 should be provided for operating costs on an annual basis. Given that, in this variant, the operation of the existing WWTP is planned during the tourist season, it is necessary to include the cost of its reconstruction, as well as the operating costs in the analysis. They amount to EUR 2,000,000 for reconstruction and EUR 101,500 per year for operating costs. Therefore, for this variant, the total investment costs are EUR 3,456,000, and the total operating costs are EUR 170,650.

The costs for the construction of a new SBR for 2000 equivalent inhabitants and the reconstruction of the existing WWTP are shown in Table 7. According to the construction dynamics plan, the construction would take place in four phases: construction of the SBR system for 2000 equivalent inhabitants with all the elements necessary for pretreatment and biological treatment (phase 1), reconstruction of the existing WWTP (phase 2), construction of a press (phase 3) and construction of a dryer (phase 4).

**Table 7.** Construction of new SBR for 2000 equivalent inhabitants and reconstruction of the existing WWTP.

| SBR system for 2000 equivalent inhabitants | | |
|---|---|---|
| Dynamics of construction by phases | Costs | Amount |
| / | Investment costs | [EUR] |
| Phase 1 | Construction of a WWTP with all the elements necessary for pretreatment and biological treatment | 806,000 |
| Phase 3 | Construction of a sludge dewatering press | 300,000 |
| Phase 4 | Construction of a dryer for drying sludge | 350,000 |
| | Σ | 1,450,500 |
| | Operating costs | [EUR/year] |
| | Electricity consumption<br>- On average, 0.90 kWh/m$^3$ of purified water and for regular equipment service | 13,500 |
| | Ongoing service staff | 7300 |
| | Maintenance of the dryer—on average, 23.77 kWh/ton of sludge produced | 47,110 |
| | Maintenance of the press—on average, 12.5 kWh (press would operate 2–3 h per day) | 1240 |
| | Σ | 69,150 |
| Existing WWTP "Toplec" | | |
| Dynamics of construction by phases | Costs | Amount (excluding VAT) |
| | Investment costs | [EUR] |
| Phase 2 | Reconstruction of the station | 2,000,000 |
| | Σ | 2,000,000 |
| | Operating costs | [EUR/year] |
| | Electricity consumption and regular equipment servicing | 71,500 |
| | Ongoing service staff | 30,000 |
| | Σ | 101,500 |

### 3.3. Variant 3—Construction of a New MBBR Wastewater Treatment System for 6000 Equivalent Inhabitants

The third variant is essentially the construction of a new MBBR wastewater treatment system for 6000 equivalent inhabitants with two modules, where both modules will be active in the tourist season and only one outside the tourist season. The location of the new station is planned next to the existing one, on part of the parcel KP 295. The area it would occupy is approximately 2400 m², and it should be subject to expropriation. Although this variant does not assume the operation of the existing station, it is planned to place a distribution shaft with a gate valve on the collector of the last pumping station, leaving a possibility to activate the existing station, if necessary. The wastewater will be directed from the separation shaft to the newly planned treatment station, as presented in Figure 3.

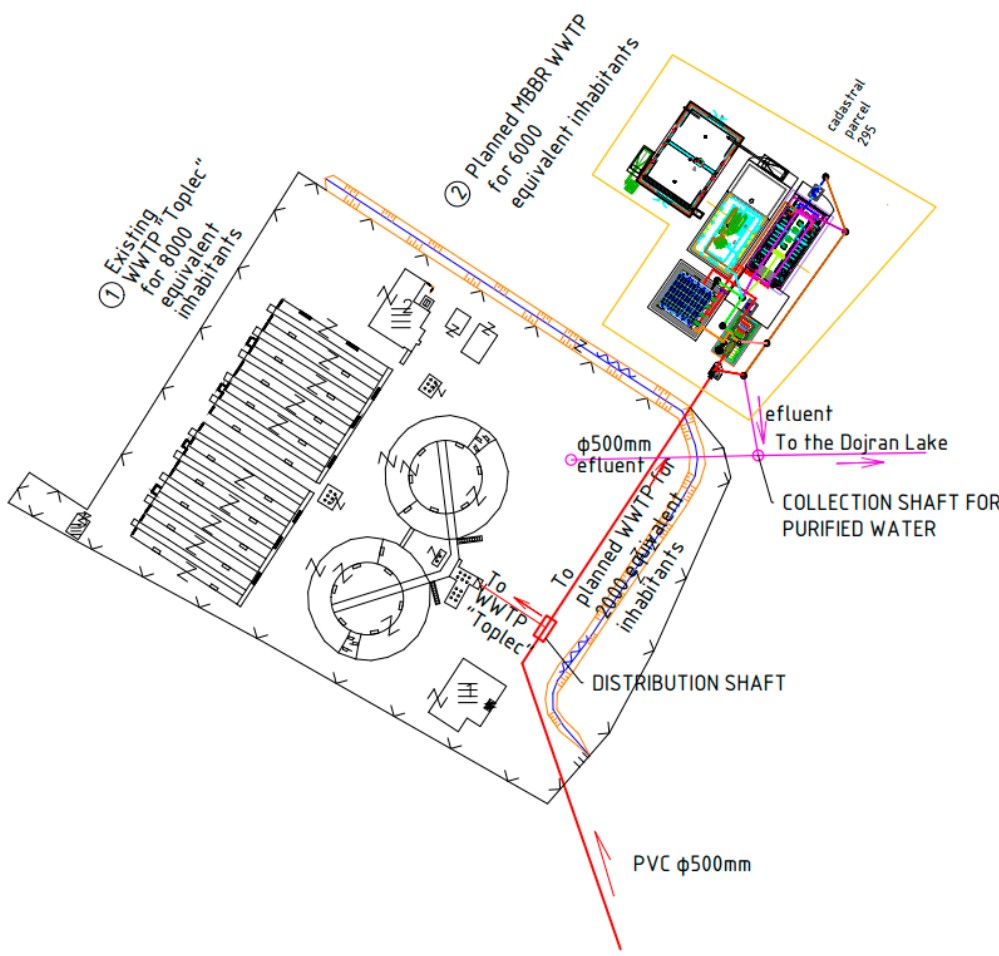

**Figure 3.** Construction of a new MBBR.

The elements, functioning and efficiency, as well as the advantages and disadvantages of the planned MBBR wastewater treatment system have already been presented in the description of Variant 1.

As for the previous two variants, the cost for the complete construction of the new MBBR wastewater treatment system for 6000 equivalent inhabitants was calculated. In addition, the costs of station maintenance and management were also calculated. According to the current market prices, the investment cost for the construction of a wastewater treatment station for 6000 equivalent inhabitants with an MBBR reactor and all the necessary stages of wastewater treatment is estimated at EUR 3,050,000 (excluding VAT). Additionally, EUR 88,150 should be provided for operating costs on an annual basis. The costs for the construction of a new MBBR system for 6000 equivalent inhabitants and the planned construction dynamics plan are given in Table 8.

**Table 8.** Construction of a new MBBR wastewater treatment system for 6000 equivalent inhabitants.

| Dynamics of construction by phases | Costs | Amount |
|---|---|---|
| / | Investment costs | [EUR] |
| Phase 1 | Construction of a WWTP with all the elements necessary for pretreatment and biological treatment | 2,400,000 |
| Phase 3 | Construction of a sludge dewatering press | 300,000 |
| Phase 4 | Construction of a dryer for drying sludge | 350,000 |
| | Σ | 3,050,000 |
| | Operating costs | [EUR/year] |
| | Electricity consumption - On average, 0.83 kWh/m$^3$ of purified water and for regular equipment service | 29,800 |
| | Ongoing service staff | 10,000 |
| | Maintenance of the dryer—on average, 23.77 kWh/ton of sludge produced | 47,110 |
| | Maintenance of the press—on average, 12.5 kWh (press would operate 2–3 h per day) | 1240 |
| | Σ | 88,150 |

## 4. Results: Selection of the Optimal Variant Using the AHP Method

In order to select the optimal technology for wastewater treatment, a complex analysis was carried out, taking into account several factors, namely, the initial investment, operating costs and complexity of the facility and equipment, as well as the need for qualified staff to operate the station. As mentioned in Section 2, we have chosen the analytic hierarchy process (AHP) method as the optimal variant of wastewater treatment process. In this specific case (Figure 4), the AHP method was implemented in the following stages:

- Setting the target function: selection of a variant solution for wastewater treatment;
- Defining decision-making criteria: initial investment, operating costs, management complexity;
- Selection of alternatives that achieve the target function: the existing WWTP and a new MBBR wastewater treatment system for 2000 equivalent inhabitants (V1), the existing WWTP and a new SBR system for 2000 equivalent inhabitants (V2), a new MBBR wastewater treatment system for 6000 equivalent inhabitants (V3).

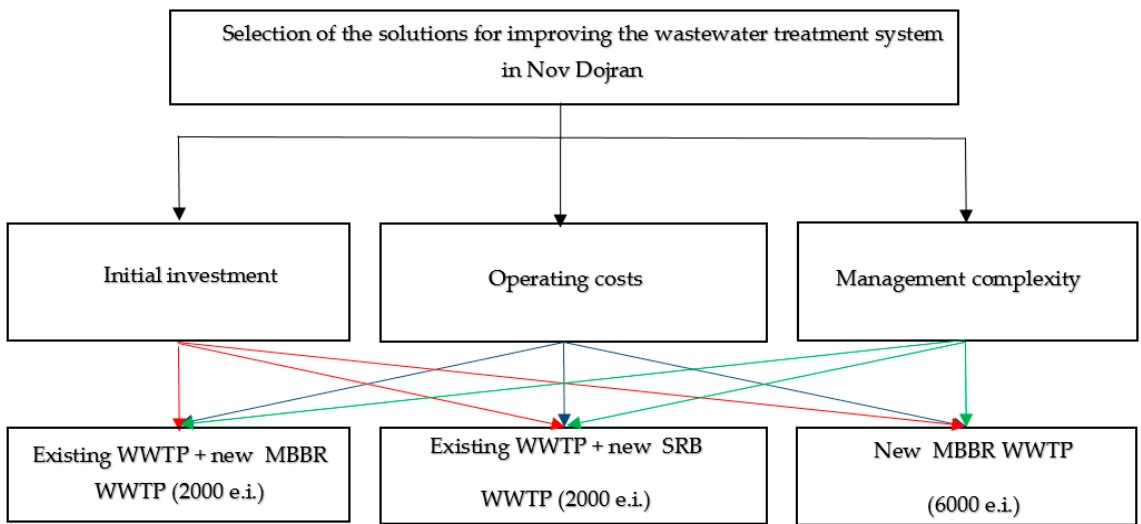

**Figure 4.** AHP scheme for the specific case.

The weights obtained after normalizing the pairwise comparison of the criteria and the weights obtained after normalizing the pairwise comparison of the alternatives/variants are shown below.

1. Criterion—(lowest) initial investment. When the variants are compared on the basis of this criterion, the sum of the investment costs of each variant appears as the value of the variant. For the first two variants, in addition to the investment cost, the amount required for the reconstruction of the existing WWTP was added.

V1—EUR 3,450,500
V2—EUR 3,456,000
V3—EUR 3,050,000
→ the value of the criteria (in EUR)
► Generation of a comparison matrix

| Variant | V1 | V2 | V3 | Sum | Weighting Coefficient Average Value |
|---------|------|------|------|-------|-------------------------------------|
| V1 | 1.00 | 2.00 | 0.25 | 3.25 | 0.23 |
| V2 | 0.50 | 1.00 | 0.25 | 1.75 | 0.13 |
| V3 | 4.00 | 4.00 | 1.00 | 9.00 | 0.64 |
| Sum | 5.50 | 7.00 | 1.50 | 14.00 | 1.00 |

► Generation of an induced matrix (normalisation)

| Variant | V1 | V2 | V3 | Sum | Weighting Coefficient Average Value |
|---------|------|------|------|------|-------------------------------------|
| V1 | 0.18 | 0.29 | 0.17 | 0.63 | 0.21 |
| V2 | 0.09 | 0.14 | 0.17 | 0.40 | 0.13 |
| V3 | 0.73 | 0.57 | 0.67 | 1.97 | 0.66 |
| Sum | 1.00 | 1.00 | 1.00 | 3.00 | 1.00 |

The weighting coefficient of the initial investment by variants is shown in Figure 5.

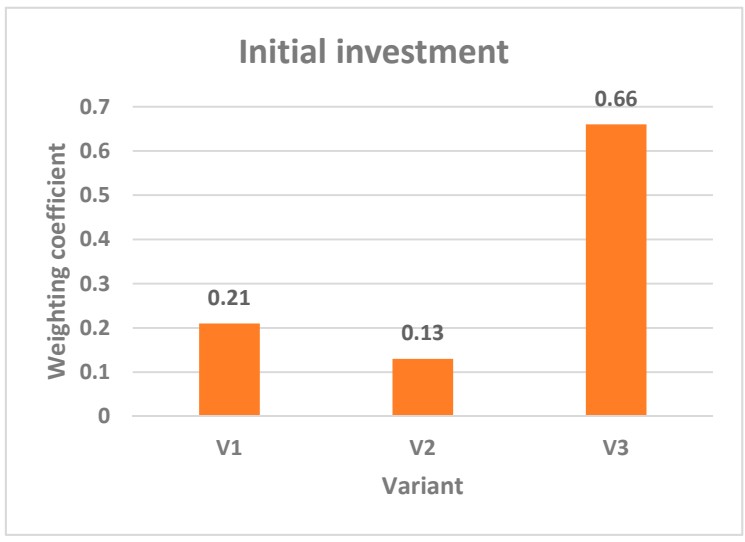

**Figure 5.** Weighting coefficient of initial investment by variants.

2. Criterion—(lowest) operating costs. In this case, the variants are compared on the basis of the resources required for the maintenance and management of the wastewater treatment system.

V1—166,250 EUR/year
V2—170,650 EUR/year

V3—88,150 EUR/year

→ the value of the criteria (in EUR/year)

► Generation of a comparison matrix

| Variant | V1 | V2 | V3 | Sum | Weighting Coefficient Average Value |
|---|---|---|---|---|---|
| V1 | 1.00 | 2.00 | 0.20 | 3.20 | 0.20 |
| V2 | 0.50 | 1.00 | 0.20 | 1.70 | 0.11 |
| V3 | 5.00 | 5.00 | 1.00 | 11.00 | 0.69 |
| Sum | 6.50 | 8.00 | 1.40 | 15.90 | 1.00 |

► Generation of an induced matrix (normalisation)

| Variant | V1 | V2 | V3 | Sum | Weighting Coefficient Average Value |
|---|---|---|---|---|---|
| V1 | 0.15 | 0.25 | 0.14 | 0.55 | 0.18 |
| V2 | 0.08 | 0.13 | 0.14 | 0.34 | 0.11 |
| V3 | 0.77 | 0.63 | 0.71 | 2.11 | 0.70 |
| Sum | 1.00 | 1.00 | 1.00 | 3.00 | 1.00 |

The weighting coefficient of the operating cost by variants is shown in Figure 6.

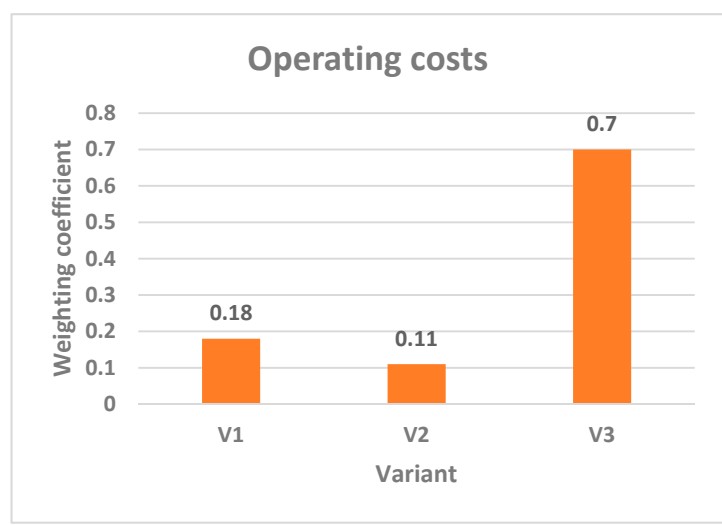

**Figure 6.** Weighting coefficient of operating costs by variants.

3. Criterion—(lowest) management complexity. In this case, the variants are compared based on the complexity of the system management, for example, starting the system operation in the tourist season and shutting it down at the end of the season, adding the medium that performs biological purification etc.

► Generation of a comparison matrix

| Variant | V1 | V2 | V3 | Suma | Weighting Coefficient Average Value |
|---|---|---|---|---|---|
| V1 | 1.00 | 0.50 | 0.14 | 1.64 | 0.08 |
| V2 | 2.00 | 1.00 | 0.14 | 3.14 | 0.16 |
| V3 | 7.00 | 7.00 | 1.00 | 15.00 | 0.76 |
| Sum | 10.00 | 8.50 | 1.29 | 19.79 | 1.00 |

► Generation of an induced matrix (normalization)

| Variant | V1 | V2 | V3 | Sum | Weighting Coefficient Average Value |
|---|---|---|---|---|---|
| V1 | 0.10 | 0.06 | 0.11 | 0.27 | 0.09 |
| V2 | 0.20 | 0.12 | 0.11 | 0.43 | 0.14 |
| V3 | 0.70 | 0.82 | 0.78 | 2.30 | 0.77 |
| Sum | 1.00 | 1.00 | 1.00 | 3.00 | 1.00 |

The weighting coefficient of management complexity by variants is shown in Figure 7.

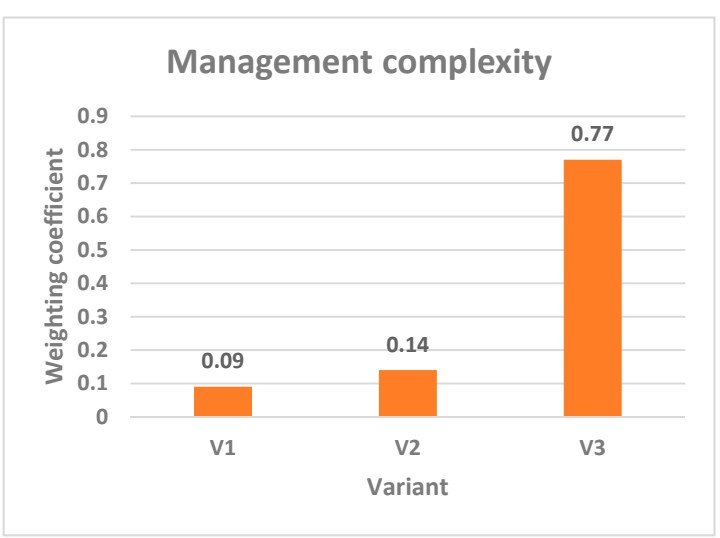

**Figure 7.** Weighting coefficient of management complexity by variants.

4. Defining the importance of the criteria

► Generation of a comparison matrix

| Criterium | C1—Initial Investment | C2—Operating Costs | C3—Management Complexity | Sum | Weighting Coefficient Average Value |
|---|---|---|---|---|---|
| C1 | 1.00 | 0.50 | 2.00 | 3.50 | 0.31 |
| C2 | 2.00 | 1.00 | 3.00 | 6.00 | 0.53 |
| C3 | 0.50 | 0.33 | 1.00 | 1.83 | 0.16 |
| Sum | 3.50 | 1.83 | 6.00 | 11.33 | 1.00 |

► Generation of an induced matrix (normalisation)

| Criterium | C1—Initial Investment | C2—Operating Costs | C3—Management Complexity | Sum | Weighting Coefficient Average Value |
|---|---|---|---|---|---|
| C1 | 0.29 | 0.27 | 0.33 | 0.89 | 0.30 |
| C2 | 0.57 | 0.55 | 0.50 | 1.62 | 0.54 |
| C3 | 0.14 | 0.18 | 0.17 | 0.49 | 0.16 |
| Sum | 1.00 | 1.00 | 1.00 | 3.00 | 1.00 |

For the purpose of greater visibility, the weighting coefficients of the criteria are shown in Figure 8.

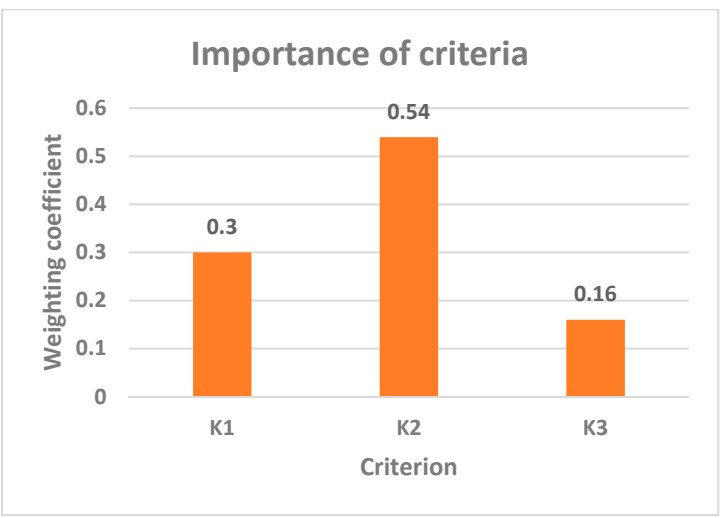

**Figure 8.** Criterion importance.

The value presented in Table 9 shows that of the three considered criteria, criterion K2—operating costs has the highest weighting coefficient, and therefore, with 54% intensity, has the greatest impact on the ranking decision. Criterion K1—initial investment with 30% intensity is in second place, while Criterion K3—management complexity, with only 16% intensity, is in the third place.

**Table 9.** Calculation with weighting coefficient.

| Criterium | C1—Initial Investment | C2—Operating Costs | C3—Management Complexity |
|---|---|---|---|
| Weighting coefficient | 0.30 | 0.54 | 0.16 |

The total priorities of the individual variants were determined so that the priorities of the variants according to each criterion (Table 10) were multiplied by the weights of the criteria (Table 9). These are shown in Table 11 and Figure 9.

**Table 10.** Priorities of variants according to each criterion.

| Variants | Total Priorities of Individual Variants | | |
|---|---|---|---|
| V1 | 0.21 | 0.18 | 0.09 |
| V2 | 0.13 | 0.11 | 0.14 |
| V3 | 0.66 | 0.70 | 0.77 |

**Table 11.** Total priorities of variants.

| Criterium/Variant | C1—Initial Investment | C2—Operating Costs | C3—Management Complexity | Sum | Sum % | Ranking |
|---|---|---|---|---|---|---|
| V1 | 0.06 | 0.10 | 0.01 | 0.18 | 17.58 | 2.00 |
| V2 | 0.04 | 0.06 | 0.02 | 0.13 | 12.50 | 3.00 |
| V3 | 0.19 | 0.38 | 0.13 | 0.70 | 69.92 | 1.00 |
| | | | $\Sigma$ | 1.00 | 100.00 | |

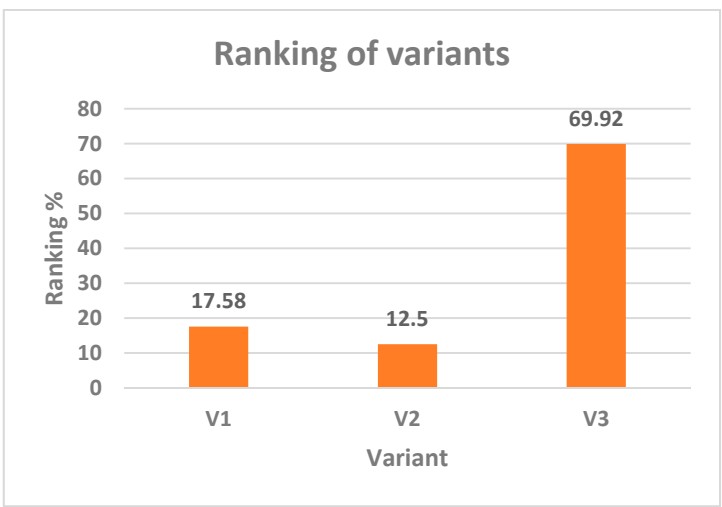

**Figure 9.** Total priorities of variants—ranking of variants.

The total priorities of the individual variants are shown in Figure 9.

According to the presented analysis, using the AHP methodology, the ranking of the variant solutions is as follows:

- Variant 1 (existing WWTP and new MBBR for 2000 equivalent inhabitants)—17.58%
- Variant 2 (existing WWTP and new SBR for 2000 equivalent inhabitants)—12.50%
- Variant 3 (new MBBR for 6000 equivalent inhabitants)—69.92%.

It can be concluded, based on the obtained results, that the AHP method suggests variant 3 as the best in all considered cases or ranking. This is a consequence of the "crossing" of the ranking of variants according to the physical characteristics of the considered parameters, independent of the ranking of the parameters according to the decision-maker's preferences. Based on the results presented above, it is obvious that Variant 3 is ahead of Variant 1 and Variant 2 according to all three selected criteria.

Through comparative analysis, it can be determined that the obtained results are consistent with the results of other research studies. Namely, the results of both older and more recent research indicate the numerous advantages of using MBBR technology in the wastewater treatment process. Yang et al. [41] concluded in their study that MBBR technology is an economically more attractive option than others, pointing to the significant savings of the capital expenditures (CAPEX). In addition to other advantages, cost savings with MBBR technology are also suggested by the results of other numerous studies [18–20], as well as its simplicity, flexibility, robustness and compactness [21].

**5. Conclusions**

In order to optimize the wastewater treatment process in the area of the municipality of Dojran, and based on the obtained results, it is recommended to design Variant 3, that is, the construction of a completely new system with MBBR technology for 6000 equivalent inhabitants. According to the AHP method, the best option is Variant 3—the construction of a new MBBR for 6000—which achieved the highest rank with 69.92%. The second-best option is Variant 1—a combination of the existing WWTP and a new MBBR for 2000—with 17.58%. Variant 2—a combination of the existing WWTP and a new SBR for 2000—achieved the lowest rank with 12.50%.

Certain limitations in the study may be partly related to the narrower selection of criteria used to select a variant solution for wastewater treatment. The limitations of the study can be partly attributed to the limitations of the application of the AHP method itself, which have already been discussed in the literature [80–82]. Therefore, the results obtained using the AHP method in this paper can be compared with the results of other methods, both in the assessment of the weight coefficients and in their use in terms of

selection, ranking and preference results. Nevertheless, we believe that the mentioned limitations cannot call into question the results of the research, but rather serve as a catalyst for future research.

**Author Contributions:** Conceptualization, J.Ć., M.K. and R.V.; methodology, J.Ć., M.K. and R.V.; validation, J.Ć, E.T. and M.G.; formal analysis, J.Ć. and E.T.; data curation, E.T. and M.G.; writing—original draft preparation, J.Ć., M.K. and R.V.; writing—review and editing, J.Ć., E.T. and M.G.; visualization, E.T. and M.G.; supervision, J.Ć. and M.K. All authors have read and agreed to the published version of the manuscript.

**Funding:** This research received no external funding.

**Data Availability Statement:** Data may be made available on request from the corresponding author.

**Conflicts of Interest:** The authors declare no conflict of interest.

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
