# Peer review of "Selection of Wastewater Treatment Technology: AHP Method in Multi-Criteria Decision Making"

_water, doi:10.3390/w15091645_

Round 1

Reviewer 1 Report

General Comments

Some times the author use “wastewater” otherwise uses “waste water”

  • Title  The title is inappropriate, but it turns out to be consistent with the article, since before the study began the results were already anticipated.
  • Keywords OK 
  • Motivation for the paper No information about the multiple-criteria decision or Analytic Hierarchy process are provided in the state-of-the-art
  • Objectives OK
  •  

THEME OPPORTUNITY –

The paper presents new perspectives of technology selection, based on Analytic Hierarchy Process. However, the paper is confusing. 

INTRODUTION –

The introduction only discusses membrane-based wastewater treatment processes, and does not mention any other type of treatment, including SBRs that later emerge as a treatment option.

In opposition, at the beginning of the paper appears a chapter dedicated to the characterization of the case study, which could integrate the material and methods. This chapter is too long and should be summarized.

METHODOLOGY-

This chapter begins with a literature review on wastewater treatment processes, in general, referring to the type of microorganisms, the importance of O2 and nutrients...

This information, besides presenting serious errors in several concepts, is totally dispensable in the "Material and Methods". It was expected that in point 3.1 "Characterization of municipal wastewater" the characterization of this wastewater would be shown, since the wastewater treatment plant already exists, as well as the treatment efficiency of this system. This information would be valuable for the reader to understand the rehabilitation of the treatment plant.

3.4 is Ok. The best section concerns the calculations of the equivalent inhabitants in the two seasons.

Chapter 4 describes the 3 variants for the treatment system. These descriptions are very detailed, so a representative figure for each option would be of more interest to the reader. Also in this chapter it is not understandable why these options were chosen and not others. A multi-criteria analysis can integrate more options than 3. A small number of variants do not require any multi-criteria analysis. Before reaching chapter 5 (which has the same title as chapter 4, it must be an oversight) the reader already knows the option.

Author Response

The authors would like to thank you for your valuable comments, advices and recommendations that have allowed us to improve the quality of the paper. We completely respect your opinion on the working version of our paper. Based on Reviewer 1 comments the entire paper has undergone certain changes, we have accepted majority of the comments and significantly improved the quality of our manuscript. If there are any comments and suggestions that were not completely solved, please do not hesitate to inform us. We sincerely hope that the revised version of the manuscript will deserve your positive opinion and recommendation of the paper for publication in the reputable journal Water.

We are very thankful to you for professional organization and quick review process. The paper has been revised carefully based on the comments from the Reviewer 1 and we have marked the changes by using the ”Track changes” in the revised paper.

Reviewer 2 Report

The paper is a very useful example of what this reviewer describes as "a how to do it" manuscript and would certainly be of value to those undertaking design of relatively small process plants in the water treatment field. 

The first language of the authors is not English, and they are advised to obtain assistance in English expression from an English native speaker.

The paper describes the process used in deciding which of three possible options should be further considered in order to increase the wastewater treatment facilities for the village of Nov Dojran in North Macedonia (line 78). Specific information on Nov Dojran is given later in the paper and the authors may wish to rearrange the way the information is given.

A small number of spelling errors were detected which may have been introduced by spelling predictive software. Such errors need to be corrected. While not complete, this reviewer noted: lines 111 and 119 "lake" rather than "Lake". Table 7 "od" rather than "of" and line 213 "Inornanic" rather than "Inorganic". And in line 595, "staff" appears as "stuff!

On line 281, reference is made to the North Macedonian denar. However, in the comparison tables towards the end of the paper, costs are given in Euros. It is recommended that the reference to denars be given in Euros say with a conversion rate of 1 denar to 0.016 Euros.

On line 140, the pipe diameter was goven as "250 m". Should it not be "250 mm"?

Author Response

The authors would like to thank you for your valuable comments, advices and recommendations that have allowed us to improve the quality of the paper. We completely respect your opinion on the working version of our paper. Based on Reviewer 2 comments the entire paper has undergone certain changes, we have accepted all the comments and significantly improved the quality of our manuscript. If there are any comments and suggestions that were not completely solved, please do not hesitate to inform us. We sincerely hope that the revised version of the manuscript will deserve your positive opinion and recommendation of the paper for publication in the reputable journal Water.

We are very thankful to you for professional organization and quick review process. The paper has been revised carefully based on the comments from the Reviewer 2 and we have marked the changes by using the ”Track changes“ in the revised paper.

Reviewer 3 Report

Manuscript water-2280633 entitled “MBBR Wastewater Treatment: AHP Method in Multi-Criteria Decision Making”. 

Please notice the following:

General view: The manuscript illustrated a great idea to use of  moving bed biofilm technology in wastewater treatment plants. The topic is very interesting.This problem is relevant for journal scope. 

The Autors expressed their idea in moderate language and grammar. The manuscript might require copyediting and proofreading up to a little degree to provide more simplified sentences.

The introduction is easy detailed. The concept and aim are clearly defined.  The presentation and discussion of the presented topicis clear and very detailed. 

Suggests supplementing the "Introduction" with information bringing a new scientific contribution. Please provide a precise review of the literature in this area.The text can be supplemented with information on the use of new carriers in MBBR technology in WWTPs.

To raise the level of paper, please use the articles on the efficiency of MBBR system in wastewater treatment plants:

doi: 10.18231/j.ijmr.2022.027

doi: 10.1016/j.biortech.2015.08.020

doi: 10.1016/j.jwpe.2022.103224

In general, the paper is well written, the results are conclusive and of interest for this topic. I have not found any important formal mistakes or typo errors. Formally, the paper is well written and easy to understand.

Please cite more papers from MDPI journals at the last 2-3 years in the similar topic of this research.

Other weaknesses to be corrected:

1. Keywords should be in alphabetical order.

2. Abbreviations not explained in the text? e.g. COD, HRT, please complete in the text

3. Correct indexes in BOD5, km2, etc (e.g. line 97, 112, 113, 225, 226, table 2)

4. There is "wastewater" in the text, it should be "wastewater" - correct it

5. The conclusions should be shorter and more specific. I propose to shorten the conclusions and keep the most important ones. It is best to list 3-4 critical conclusions.

The manuscript follows the formal regulations of MDPI journals.

I suggest the acceptance after minor revision

Author Response

The authors would like to thank you for your valuable comments, advices and recommendations that have allowed us to improve the quality of the paper. We completely respect your opinion on the working version of our paper. Based on Reviewer 3 comments the entire paper has undergone certain changes, we have accepted all the comments and significantly improved the quality of our manuscript. If there are any comments and suggestions that were not completely solved, please do not hesitate to inform us. We sincerely hope that the revised version of the manuscript will deserve your positive opinion and recommendation of the paper for publication in the reputable journal Water.

We are very thankful to you for professional organization and quick review process. The paper has been revised carefully based on the comments from the Reviewer 3 and we have marked the changes by using the ”Track changes“ in the revised paper.

Round 2

Reviewer 1 Report

General Comments

  • Title  has been correctly improved
  • Keywords OK 
  • Objectives OK
  • Conclusions OK 

THEME OPPORTUNITY –

The paper presents new perspectives of technology selection, based on Analytic Hierarchy Process. The article was improved. 

INTRODUTION –

The introduction, although improved, does not describe in the same detail the technologies that are used in the WWTP variants. The authors have added technologies in the introduction but make no mention of SBRs before page 12.

METHODOLOGY-

The characterization of the case study was introduced in this section.

Also, the literature review about wastewater treatment processes, was correctly removed from this chapter.

However, it was expected that the "Characterization of municipal wastewater" would be shown, since the wastewater treatment plant already exists, as well as the treatment efficiency of this system. This information would be valuable for the reader to understand the rehabilitation of the treatment plant.

Chapter 4 describes the 3 variants for the treatment system. These descriptions are very detailed, so a representative figure for each option would be of more interest to the reader. Also, in this chapter it is not understandable why these options were chosen and not others. A multi-criteria analysis can integrate more options than 3. A small number of variants do not require any multi-criteria analysis. Before reaching chapter 5 (which has the same title as chapter 4, it must be an oversight) the reader already knows the option.

In short, the article is of average quality. Thus, I am of the opinion that the article should be improved to be suitable for publication.

Author Response

We are very thankful to you for professional organization and quick review process. We are especially grateful for the constructive comments and suggestions of reviewers, which helped us a lot in improving the quality of our paper. The paper has been revised carefully based on the comments from the all reviewers and we have marked the changes by using the ‚‚Track changes“ in the revised paper.

We would like to express our sincere gratitude to the Reviewer 1 for your positive comment. We completely respect your opinion on the working version of our paper. Based on reviewers' comments the entire paper has undergone changes, we have accepted all the comments and significantly improved the quality of our manuscript. We sincerely hope that the revised version of the manuscript will deserve your positive opinion and recommendation of the paper for publication in the reputable journal Water.

Round 3

Reviewer 1 Report

General Comments

  • Title  OK
  • Keywords OK 
  • Objectives OK
  • Conclusions OK 

THEME OPPORTUNITY –

The paper presents new perspectives of technology selection, based on Analytic Hierarchy Process.

INTRODUTION –

Line 117 Quan and Gogina the data ie missing

Line121 “so-cio-economic” change to “socio-economic”

Line 126 Alagha et al,. the data is missing

Line 130 Fernandes et al. data is missing

Line 131 “limited aeration i potvrduju????? viability”

CHAPTER 3- ANALYSIS

Chapter 3 describes the 3 variants for the treatment system. A representative figure was introduced, as required.

Thus, I am of the opinion that the paper is suitable for publication, after minor revision.

Author Response

(The authors gave the same response as above.)
